# The Usefulness of an Online Simplified Screening Questionnaire (SSQ) in Identifying Work-Related Cancers

**DOI:** 10.3390/healthcare11111563

**Published:** 2023-05-26

**Authors:** Fabiana L. Vazquez, Henrique C. S. Silveira, Ubirani B. Otero, Thais T. Hosokawa, José Humberto T. G. Fregnani, Adhemar Longatto-Filho, Rui M. Reis

**Affiliations:** 1Molecular Oncology Research Center, Barretos Cancer Hospital, Barretos 14784-390, SP, Brazil; 2Brazilian National Cancer Institute—INCA, Rio de Janeiro 20230-240, RJ, Brazil; 3Albert Einstein Israelita Hospital, São Paulo 05652-900, SP, Brazil; 4A.C.Camargo Cancer Center, São Paulo 01509-010, SP, Brazil; 5Faculty of Medicine, Department of Pathology, University of São Paulo, São Paulo 05508-900, SP, Brazil; 6Life and Health Sciences Research Institute (ICVS), Medical School, University of Minho, 4710-057 Braga, Portugal; 7ICVS/3B’s—PT Government Associate Laboratory, 4805-017 Guimarães, Portugal

**Keywords:** occupational cancer, environment, neoplasm, questionnaires

## Abstract

To obtain a history of occupational exposure in the workplace, the questionnaire is one of the main sources of information. The aim of this study was to develop an online questionnaire using the REDCap data management platform based on the Work-Related Cancer Surveillance Guidelines, reported by the Brazilian National Cancer Institute. Several issues were taken into consideration for its routine application. It should be simple, easy, capable of being applied in a short time and used in the clinical setting of collecting information on the occupational history of the cancer patient. Consequently, this could enable the compulsory notification of work-related cancer. The questionnaire was developed based on questions about the use of and exposure to carcinogenic factors at work and due to smoking. An entirely electronic version of the cancer patient interview was performed using tablets. The online questionnaire was applied at the Barretos Cancer Hospital, Barretos, to newly diagnosed patients from July 2016 to 2018. A total of 1063 patients were included, and 550 indicated positively when asked “Do you work, or have you worked with this substance and/or in this function?/job?” Of these potentially notified patients, 38 subsequently had compulsorily reported work-related cancer. Another important result of this study was the creation and development of a website. In conclusion, we developed an online tool that could facilitate hospital routines, contributing to generating data for the compulsory notification of work-related cancer and triggering investigations and surveillance actions in Brazil.

## 1. Introduction

All work-related diseases are generally of great relevance because of their social–cultural impact as far as public health issues are concerned, as recognized by World Health Organization (WHO) experts [1]; for the WHO, occupational exposure is, at present, the main form of exposure to carcinogens classified by the IARC (International Agency for Research on Cancer) [2]. To reiterate, this issue has been noted by World Health Organization (WHO) experts [1], and 4% of cancer cases have been attributed to occupational exposures, and it is estimated that, in industrialized countries, approximately 9% of cancers in men are due to this exposure.

The etiology of cancer is associated with environmental factors in approximately 80% of cases; of these, 30% are related to smoking, 35% to food, 10% to infections, 7% to high-risk sexual behavior and 3% to alcohol abuse; according to the pivotal work of the prestigious 1981 US study by Doll et al. [3] and up to now, these figures have not changed substantially. Brazilian data, however, have shown a rate of 1.56 cases of work-related cancer per 100,000 insured workers with in the social security program in 2013, and presumably, this is due to the lack of mandatory notifications of cases [4]. To illustrate the point, for lung cancer alone, European statistics predict that 1in 10cases of cancer may be due to work, which would be approximately 1780 cases of lung cancer in Brazil in 2010 [4]. There are significant work-related cancers with rates of 10.44 cases per 100,000 insured workers in France, 9.86 in Belgium, 6.57 in Germany, 6.53 in Finland and 5.15 in Italy [5].

### Brazilian Work-Related Cancer Surveillance Guidelines

Briefly, the Work-Related Cancer Surveillance Guidelines were published in Brazil in 2012 and revised in 2013 by the Brazilian National Cancer Institute [6]. These guidelines were elaborated with the intention of making a technical contribution to addressing cancer- and carcinogen-related risks present in the workplace. Furthermore, they were developed by consensus with international guidelines to control the potential exposure to carcinogens in different types of workplaces and prevent their possible effects on human health.

Moreover, the guidelines aim to provide technical and epidemiological regulations that favor workers’ professional activities and make it easier to obtain detailed information about potential carcinogens present in work environments [6].

Finally, the purpose of the Guidelines is to facilitate the action of integrated care and prevention by sharing responsibilities, information and instruments with society, organized movements and other groups directly involved in these fields, such as educational groups, as well as environmental and developmental policies [6].

Paradoxically, in reality, data on work-related cases of cancer in Brazil are extremely poor and scarce, and health professionals know little or nothing about work processes and carcinogens present in work environments; consequently, there is no approach to recording the history of labor occupation in anamneses because health professionals are unaware of these processes.

The non-representativeness of work-related cancer in Brazil can be established based on occupational accident records (the Work Accident Report—CAT), in which cancer cases in 2009 accounted for only 0.23% of cases of occupational diseases according to INCA [7].

Therefore, the following should be considered:i  In Brazil, there are Work-Related Cancer Guidelines that have been elaborated with clear and notable aims.ii There is an information system regulating compulsory notification offenses, and Brazil has adopted a form for registering information and notifications, neither of which are being well used or implemented.iiiAlthough provided by the INCA manual, this form has never been implemented in clinical practice, possibly because of the difficulty in establishing a clear link between disease and occupational risk factors.

Based on all the foregoing considerations, this study was designed to determine the applicability of these variables and seek solutions to make it feasible to obtain information in real scenarios of cases of malignant neoplasms diagnosed at Barretos Cancer Hospital, which would help identify exposures that occurred in workplaces.

Based on the above-mentioned INCA Guidelines, a practical and simplified questionnaire was developed. It is simple, easy and quick to apply (only available in the electronic version for smart phones or tablets) in meeting the objectives of obtaining information on exposure and occupation and verifying the feasibility of using it in a clinical setting to obtain information for purposes of epidemiological registration and the possible compulsory notification of work-related cancer.

## 2. Materials and Methods

### 2.1. Development of the Simplified Screening Questionnaire (SSQ) for Work-Related Cancer

The SSQ arose from the need to obtain epidemiological data on carcinogenic exposure which cancer patients a tour institution might have experienced in the workplace. Starting from this point, we searched for information in the international and national literature and the Brazilian Work-Related Cancer Surveillance Guidelines.

We chose the Brazilian Guidelines because they are a national guide, written for the same purpose as our study. However, as far as we know, they have never been put into practice, and no results have been published so far.

We found that, in clinical practice, it would be almost impossible to apply this entire body of information without specific training and without a systematic collection tool that health professionals could use to obtain the necessary information from the patient’s occupational history, considering that there are many types of occupational exposure-related neoplasms and a host of carcinogenic factors that can be found in many workplaces.

Thus, a project was created to develop a tool that would make it easy, simple and quick for professionals to gather the necessary information on occupational exposure and cancer.

### 2.2. First Step

i  List all types of cancer that could be attributed to work-related exposure.ii For each type of cancer, list the carcinogens that could be present in the workplace and elsewhere.iiiFor each carcinogen, list the workplaces and/or professions in which they could be present.ivCalculate the time of exposure.v Calculate the time of latency from exposure to the onset of cancer.

The questionnaire was tailored to each type of cancer, with a list of specific carcinogens and workplaces/occupations where they might be present. The list was based on the Brazilian Guidelines for the Surveillance of Work-Related Cancer, which outlines common workplace carcinogens that can lead to the development of certain types of cancer. This approach allows researchers to gather more targeted and specific information about each participant’s exposure history and potential risk factors for cancer (Appendix A) (Figure 1).

### 2.3. Second Step

After having gathered the information shown in the lists above, we developed the questionnaire using REDCap (data management software) [8], which allowed us to create numerous logic branches and relationships and perform calculations to obtain the data we needed. Furthermore, it was possible to apply the questionnaire to cancer patients in a completely electronic version using tablets. The aim was to draw professionals’ attention directly to the main questions exclusively on exposures pertinent to the patient’s specific type of cancer only, thereby making it possible to find the information and minimize the time taken to do so.

For this, each type of cancer, with its possible relationship with occupational exposure, was grouped in the following questions (Figure 2).

### 2.4. Third Step

This was a cross-sectional prospective data collection study.

After the questionnaire was developed, it was used on cancer patients diagnosed from 2017 to July 2018.Those who wanted to participate in the study answered the questions of the SSQ.

Patients diagnosed with cancer that had a possible relationship with their occupations answered the SSQ in the following HCB departments: head and neck/skin and melanoma; high digestive; thorax; hematology; breast; urology; and neurology. The duration of the application of the questionnaire was also calculated in real time by using the REDCap project, which was developed for data collection in this study.

The Barretos Cancer Hospital is a reference center for cancer prevention and treatment in Brazil and Latin America with extensive experience in treatment, research and the prevention of cancer.

### 2.5. Study Population

Inclusion criteria were men and women, 18 years of age or older, who had worked in their lifetime and were recently diagnosed with one of the following cancers: melanoma, lung, mesothelioma, bladder, nasal cavity and paranasal sinuses, oral cavity, pharynx and larynx, leukemia, myelodysplasia, multiple myeloma, non-Hodgkin’s lymphoma, stomach and esophagus, liver, pancreas, breast, or central nervous system. Furthermore, we included those who were diagnosed during the month of recruitment, legally competent to consent and willing to participate in the study.

Exclusion criteria: Those who had never worked and refused to sign the terms of free and informed consent were not included in the study.

### 2.6. Ethics

Consent was obtained by research coordinators from each department treating patients in various sectors of the hospital, such as outpatient clinics, the infusion center, the hospitalization department, etc., in a reserved and appropriate place and always respecting patients’ consent to undergoing procedures.

The terms of consent were signed in 2 copies, 1physical (printed) and 1digital (tablets using the REDCap Mobile App). The physical copy was handed to the participant, and the digital copy was retained in study files. This project was approved by the Ethics Committee of Barretos Cancer Hospital, CAAE number: 51644015.8.0000.5437. The privacy rights of human subjects were observed.

### 2.7. Privacy, Data Storage and Confidentiality

All data are securely stored in a designated place and in protected-document storage facilities at Barretos Cancer Hospital. Participants were informed of the data storage of their Terms of Consent and assured of their confidentiality. Names of and information about study participants remained confidential and were reviewed exclusively by the principal investigator and his study coordinator only when necessary.

### 2.8. Sample Calculation

A convenience sample was considered consisting of patients in the age group of over 18 years, and new cases of cancer with a possible relationship with an occupation were included in the study. In the period from 2017 to July 2018, the number of patients screened was 1234, and of these, 1063 were included in the study.

### 2.9. Study Conducted in Steps to Enable Use of SSQ

Training and data collection were carried out prior to the beginning of the project. For convenience, we chose the department of skin cancer and melanoma to develop a pilot study and test the information system. Tablets were purchased for each of the research coordinators of each hospital department. The REDCap Mobile App was installed on the tablets so data collection could be performed offline and uploaded when connected to the internet.

## 3. Results and Discussion

The first pilot study was conducted after training the nurses of the department of skin cancer and melanoma. At the time of the first consultation screening, the nurse in charge of the work applied the terms of informed consent and the study questionnaire.

This first part of data collection occurred from February to June 2016 and resulted in 55 inclusions. Afterward, the researcher support team, coordinators and project assistants underwent training to assist with the implementation of the questionnaire. Project coordinators and assistants interviewed potential participants at, for example, outpatient clinics, infusion centers, hospitals, radiotherapy departments and radiology centers. There were 51 additional inclusions from June to July 2016.

The REDCap data collection project was adjusted to include the latency time; thus, in addition to questions about the time that patients worked in these occupational environments and occupational hazards, they also answered questions about how long it was between when they stopped working and the time the onset of cancer occurred, and thus, the latency time for each exposure was calculated.

### 3.1. Patient Recruitment

Recruitment occurred throughout the hospital with newly diagnosed patients, who were first registered or underwent exams before starting their treatment. Up to July 2018, the number of patients screened was 1262, and of these, 1063 were included in the study.

The main result of this study on the use of SSQ was that it made it possible to identify patients who were exposed to some type of carcinogen in their work environments; measure the time of application of the questionnaire; and verify the feasibility of using the questionnaire in a cancer hospital service routine.

The notifiable term was used as raw data whenever occupational exposure was reported, irrespective of exposure time and latency time.

This refers to a term used in reporting occupational exposure that requires notification to the appropriate authorities. The “notifiable term” is used regardless of the duration of exposure or the time between exposure and the onset of symptoms.

Therefore, of the 1063 patients included in the study, 550 were notifiable cases; that is, they answered with any of the alternatives indicated in the questionnaire when asked question 4: “Do you work, or have you ever worked with this substance and/or in this function?/job?”.

The time taken to apply the consent form and questionnaire averaged 7.56 min and a median of 6 min.

The general results for gender were 626 (49.6%) women and 636 (50.4%) men, with the majority of them living in the state of São Paulo (50.3%). Regarding education, the findings were as follows: 3.9% were illiterate; several had incomplete 1st-through 4th-grade elementary school education; 780 (61.8%) had less than 8 years of schooling; and 482 (38.2%) had more than 8 years of schooling. Of the 1063 patients included in the study, 435 had never smoked (42.6%), 180 were smokers (17.6%) and 406 were former smokers (39.8%).

The main reasons for ineligibility were as follows: 108 of those recruited had never worked (neither professional activity nor at home) (54.8%); 6 of them had cognitive impairment (3.0%); and 83 had other reasons (42, 1%). Among the other reasons, the main one was a type of tumor that was not related to an occupation (cervix, colon, etc.).

#### The Distribution of Participants by Medical Department

Head and neck/skin and melanoma: (314, 30.8%): non-melanoma skin (170, 54.1%), melanoma (34, 10.8%), nasal cavity and nasal sinuses (5, 1.6%), oral cavity (45, 14.3%), pharynx (36, 11.5%), larynx (24, 7.6%).Upper digestive system: (189, 18.5%): stomach (106, 55.5%), esophagus (43, 23.6%), liver (6, 3.3%), pancreas (32, 17, 6%).Thorax: (127, 12.5%): lung (115, 97.5%), mesothelioma (3, 2.5%).Hematology: (105, 10.3%): lymphoid leukemia (2, 1.9%), myeloid leukemia (13, 12.4%), myelodysplasias (2, 1.9%), multiple myeloma (18, 17, 1%), non-Hodgkin’s lymphoma (70, 66.7%).Breast: (184, 18.0%).Urology: (48, 4.7%).Neurology: (53, 5.2%).

Regarding distribution in the departments by the type of cancer and exposure to any carcinogenic agents present in the workplace, we obtained the following results (Table 1).

Regarding the distribution by type of cancer and exposure to the most common carcinogens present in the workplace, we obtained the following results in Table 2.

Agriculture, specifically, applying or handling pesticides (herbicides, pesticides, insecticides, fungicides), was the most frequent; the herbicide glyphosate and the insecticides malathion and diazinon are classified as probably carcinogenic to humans (Group 2A) by the IARC—International Agency for Research on Cancer.

Regarding the notifications generated based on the distribution by type of cancer and exposure to the most frequent carcinogens present in the workplace, we obtained the following results, shown in Table 3.

The criterion used to notify cases were non-smoker; time of exposure to the carcinogen at work; latency time; and information about the function in the workplace and the activity carried out. The cases were analyzed individually, and the minimum time of exposure and latency considered for reporting was 15 years. Up to the time of the beginning of this research, Barretos Cancer Hospital had not made any compulsory reports of work-related cancer. After this study, 10 cases of mesothelioma, 5 of lung cancer, 3 of non-Hodgkin’s lymphoma, 1 of lymphoid leukemia, 3 of multiple myeloma, 1 of cancer of the central nervous system, 2 of cancer of the bladder, 1 of breast cancer, 1 of liver cancer, 2 of cancer of the esophagus, 6 of stomach cancer, 1 of larynx cancer, 1 cancer of the pharynx and 1 cancer of the nasal cavity and paranasal sinuses were reported, totaling 38 notifications.

Another important result of this study was the creation and development of a website, which can currently be accessed at this address: qsimplificado.com.br (accessed on 22 February 2023).

### 3.2. Applicability

The SSQ was developed and applied in a research environment with the purpose of being disseminated to other institutions as a tool for collecting epidemiological data and enabling the compulsory notification of work-related cancers, thus triggering actions of investigation and surveillance. With this objective in mind, in the routines of the departments at our institution, we developed and tested a website where institutions can register and receive access to the SSQ. The data collected will be stored on a secure server at Barretos Cancer Hospital, which will manage and make these data available to the participant institutions, each accessing their own site content.

In addition to the registered institutions, our goal is also to reach the general population diagnosed with cancer by making the page available for the self-filling-out of data. Thus, in real time, we could obtain information from a national database of institutional and public sources about cancer and occupational activity, something unprecedented considering the scarcity of data in the Brazilian territory. Cancer participants will be registered by inserting their National Individual Registration number (a type of unique personal identification number issued and controlled by the Internal Revenue Department CIC), thus avoiding duplicate information on the website developed to provide other institutions with access to the SSQ (Figure 3).

The results of implementing the system to collect information about work-related cancer by using a very simplified questionnaire to collect data—which will facilitate the recognition of potential carcinogens in workplace environments—were demonstrated to be a very promising tool that will be shared with different areas throughout the Brazilian territory.

We demonstrated that important data can be efficiently collected using this instrument in clinical routines because it is simple, easy and quick to apply; moreover, it would be capable of providing information about the occupational history of the development of cancer.

Based on the first results, it was found to be feasible to obtain initial data for epidemiological records on professions and exposure to carcinogens. It was also possible to trace the profile of patients relative to smoking, education, place of residence, type of exposure and the most common profession at risk of each type of cancer and to report the first cases for registration in the Brazilian SINAN—Notification of Disease Information System.

## 4. Limitations of the Study

A limitation of this study is the high sensitivity and low specificity of the questionnaire. However, this questionnaire can help in occupational cancer surveillance regarding the main risk factors found in a population, and steps can be taken to prevent cancer, such as minimizing farmers’ exposure to agricultural chemicals.

## 5. Conclusions

Based on the premise that, among health professionals, little is known about carcinogens present in the work environment and that there has been practically no approach to recording the work history in patient anamneses, the SSQ was successfully used at our institution. The project (SSQ) achieved the objectives proposed by providing the first surprising results in our country. These results can serve as technical subsidies for prioritizing health surveillance actions, and other institutions in our country will be able to adopt this instrument in their routine work.

## Figures and Tables

**Figure 1 healthcare-11-01563-f001:**
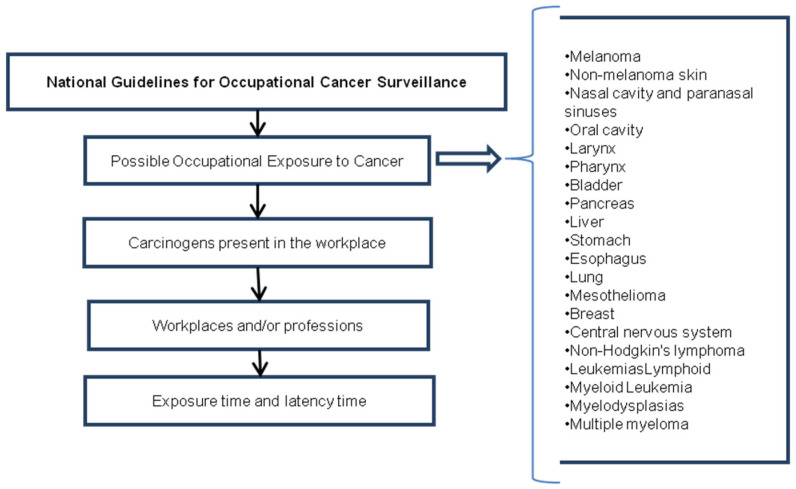
First step in building the SSQ questionnaire.

**Figure 2 healthcare-11-01563-f002:**
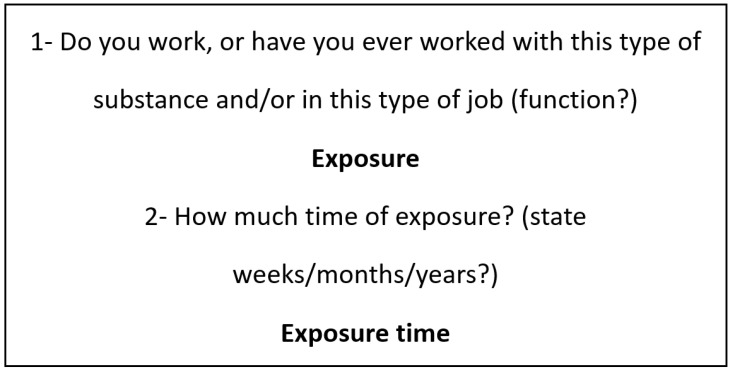
Second step—questions were asked about the relationship between the grouped types of occupational exposures with each type of cancer.

**Figure 3 healthcare-11-01563-f003:**
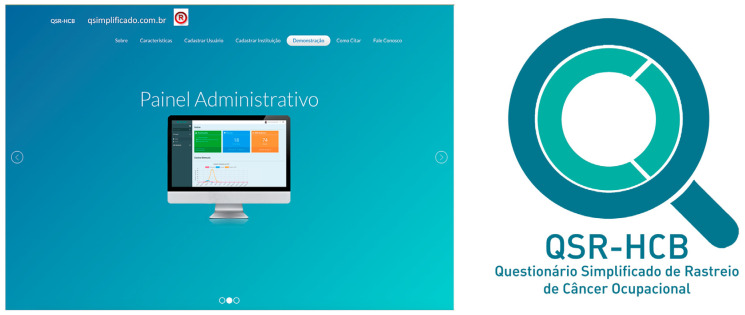
Homepage and registration of SSQ institutions—qsimplificado.com.br (accessed on 22 February 2023).

**Table 1 healthcare-11-01563-t001:** Distribution of the number of patients exposed to some type of carcinogen in the workplace among cancer patients who responded to the SSQ.

Department Head and Neck/Skin and Melanoma	*n*	*n* Exposed	% Exposed
Non-melanoma skin	170	150	88.2
Melanoma	34	23	67.6
Nasal cavity and paranasal sinuses	5	4	80.0
Oral cavity	45	32	71.1
Pharynx	36	28	77.8
Larynx	24	20	83.3
Total	314	257	81.8
**Upper Digestive Department**	** *n* **	***n* Exposed**	**% Exposed**
Stomach	106	59	55.7
Esophagus	44	27	61.4
Liver	6	5	83.3
Pancreas	33	8	24.2
Total	189	99	52.4
**Thorax Department**	** *n* **	***n* Exposed**	**% Exposed**
Lung	123	61	49.6
Mesothelioma	3	2	66.7
Total	126	63	50.0
**Hematology Department**	** *n* **	***n* Exposed**	**% Exposed**
Lymphoid leukemia	2	1	50
Myeloid Leukemia	13	6	46.2
Myelodysplasias	2	2	100.0
Multiple myeloma	18	10	55.6
Non-Hodgkin’s Lymphoma	70	34	48.6
Total	105	53	50.5
**Breast Department**	** *n* **	***n* Exposed**	**% Exposed**
Breast	184	38	20.7
**Urology Department**	** *n* **	***n* Exposed**	**% Exposed**
Bladder	48	20	41.7
**Neurology Department**	** *n* **	***n* Exposed**	**% Exposed**
Central Nervous System	53	20	37.7

**Table 2 healthcare-11-01563-t002:** Distribution of the number and percentage of cancer patients who responded to the SSQ and exposures to the most common and frequent carcinogens in the workplace.

	**SKIN—NOT MELANOMA**	*n*	%
**Exposition**	In agriculture, applying or handling pesticides (herbicides, pesticides, insecticides, fungicides)	59	39.3
	Exposed to sunlight or exposed to ultraviolet radiation in an outdoor occupation (bricklayer, welder, salesman, farm worker, lifeguard, health worker, fisherman, traffic guard, gardener, mountaineering guide, miner)	149	99.3
	**MELANOMA**	*n*	%
	Exposed to sunlight or exposed to ultraviolet radiation in an outdoor occupation (bricklayer, welder, salesman, farm worker, lifeguard, health worker, fisherman, traffic guard, gardener, mountaineering guide, miner)	21	91.3
	**NASAL SINUS OF PARANASAL SINUSES**	*n*	%
	Handling herbicides (Paraquat or Gramoxone) in agriculture	3	75
	Placement or demolition of asbestos products such as in construction; plumbing; laying and remodeling roofs (Eternit tiles); thermal insulation of boilers and pipes	3	75
	**ORAL CAVITY**	*n*	%
	In direct contact with wood dust, leather dust, cement dust, or cereal dust; in direct contact with textile dust	17	53.1
	applying or handling pesticides (herbicides, pesticides, insecticides, fungicides) in agriculture	16	50
	Placement or demolition of asbestos-based products such as construction; plumbing; boiler and piping thermal remodeling (roofing); boiler and piping thermal insulation	16	50
	Animal husbandry	14	43.8
	**PHARYNX**	*n*	%
	In direct contact with wood dust, leather dust, cement dust or cereal dust; in direct contact with textile dust	19	67.9
	Applying or handling pesticides (herbicides, pesticides, insecticides, fungicides) in agriculture	13,	46.4
	Timber, Sawmill and Joinery Industry (creosote)	12	42.9
	**LARYNX**	*n*	%
	Applying or handling pesticides (herbicides, pesticides, insecticides, fungicides) in agriculture	11	57.9
	Animal husbandry	10	52.6
	In direct contact with wood dust, leather dust, cement dust or cereal dust; in direct contact with textile dust	8	42.1
	**STOMACH**	*n*	%
	Applying or handling pesticides (herbicides, pesticides, insecticides, fungicides) in agriculture	39	66.1
	With construction dust (bricklayers, etc.)	27	45.8
	**ESOPHAGUS**	*n*	%
	Applying or handling pesticides (herbicides, pesticides, insecticides, fungicides) in agriculture	15	55.6
	With construction dust (bricklayers, etc.)	12	44.4
	Gas station attendant (or with fuel vapors)	5	18.5
	**LIVER**	*n*	%
	Applying or handling pesticides (herbicides, pesticides, insecticides, fungicides) in agriculture	3	60.0
	Copper production	1	20.0
	Wood industry with creosote used for wood preservation	1	20.0
	The production or manufacture of paints, plastics, rubber products, pigments and papers	1	20.0
	**PANCREAS**	*n*	%
	Applying or handling pesticides (herbicides, pesticides, insecticides, fungicides) in agriculture	8	100
	Wood industry with creosote used for wood preservation	1	12.5
	**LUNG**	*n*	%
	Applying or handling pesticides (herbicides, pesticides, insecticides, fungicides) in agriculture	30	49.2
	Placement or demolition of asbestos products such as in construction; plumbers; laying and remodeling roofs (Eternit tiles); thermal insulation of boilers and pipes	16	26.2
	**MESOTHELIOMA**	*n*	%
	Placement or demolition of asbestos products such as in construction; plumbers; laying and remodeling roofs (Eternit tiles); thermal insulation of boilers and pipes	2	100
	**LYMPHOID LEUKEMIA**	*n*	%
	Applying or handling pesticides (herbicides, pesticides, insecticides, fungicides) in agriculture	1	100
	**MYELOID LEUKEMIA**	*n*	%
	Applying or handling pesticides (herbicides, pesticides, insecticides, fungicides) in agriculture	3	50
	**MYELODYSPLASIA**	*n*	%
	Applying or handling pesticides (herbicides, pesticides, insecticides, fungicides) in agriculture	2	100
	Asbestos extraction mining (asbestos rock extraction processes, drilling, cutting, dismantling, crushing, screening and handling these rocks)	1	50
	**MULTIPLE MYELOMA**	*n*	%
	Applying or handling pesticides (herbicides, pesticides, insecticides, fungicides) in agriculture	7	77.8
	**NON-HODGKIN’S LYMPHOMA**	*n*	%
	Applying or handling pesticides (herbicides, pesticides, insecticides, fungicides) in agriculture	22	64.7
	**BREAST**	*n*	%
	Applying or handling pesticides (herbicides, pesticides, insecticides, fungicides) in agriculture	17	44.7
	Direct contact with formaldehyde (such as hair stylists using permanent brushes and volatile organic compounds, among other occupations that work directly with formaldehyde)	13	34.2
	**BLADDER**	*n*	%
	Applying or handling pesticides (herbicides, pesticides, insecticides, fungicides) in agriculture	12	63.2
	Exposed to diesel flue gases	7	36.8
	**CENTRAL NERVOUS SYSTEM**	*n*	%
	Applying or handling pesticides (herbicides, pesticides, insecticides, fungicides) in agriculture	8	40
	Wood industry with creosote used for wood preservation	4	20
	Metallurgy and steelmaking (contact with metal waste such as mercury, lead and cadmium); handling heavy metal waste	4	20

**Table 3 healthcare-11-01563-t003:** Distribution of the most frequent exposures to carcinogens or occupations of cancer patients in the SSQ.

Cancer	Carcinogen or Professional Occupation											
	Agriculture	Sun Exposed	Asbestos	Dust	Breeding	Mason	Gas station Attendant	Diesel	Creosote	Cooper	Formalin	Heavy Metals
Non-melanoma skin	59	149										
Melanoma		21										
Paranasal sinuses nasal cavity	3		3									
Oral cavity	16			17	14							
Pharynx	13			19					12			
Larynx	11			8	10							
Stomach	39					27						
Esophagus	15					12	5					
Liver	3								1	1		
Pancreas	8								1			
Lung	30		16									
Mesothelioma			2									
Lymphoid leukemia	1											
Myeloid leukemia	3											
Myelodysplasia	2		1									
Multiple myeloma	7											
Non-Hodgkin’s lymphoma	22											
Breast	17										13	
Bladder	12							7				
CNS	8								1			4

## Data Availability

The data presented in this study are available upon request from the corresponding author. The data are not publicly available due to the participants’ right to anonymity (confidentiality) guaranteed in the terms of consent signed by the participants and the researchers.

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
