# Peer review of "The Usefulness of an Online Simplified Screening Questionnaire (SSQ) in Identifying Work-Related Cancers"

_healthcare, 2023, doi:10.3390/healthcare11111563_

Round 1
Reviewer 1 Report
please see attachment

Author Response
Response to Reviewer 1 Comments
Point 1: Results and Discussion section mentions a total of 1,063 patients were included, and 550 indicated positive when asked “Do 27 you work, or have you worked with this substance and/or in this function? /job?”. Were the same questionnaires given to all the participants or there were modifications in the questions according to the types of cancers they reported. What was the criteria of listing the suspected carcinogens in the questionnaire. Was the list of carcinogens different for the patients with different types of cancers? Please, elaborate this part in the methods section.
Response 1: We would like to thank the reviewer comment. This part has been added to the methods section (lines 135-141):
“The questionnaire was tailored to each type of cancer, with a list of specific carcinogens and workplaces/occupations where they might be present. The list was based on the Brazilian Guidelines for Surveillance of Work-Related Cancer, which outlines common workplace carcinogens that can lead to the development of certain types of cancer. This approach allows researchers to gather more targeted and specific information about each participant's exposure history and potential risk factors for cancer.”
The questionnaire was included in supplementary material: Annex (document: SSQ_Attachment)
Point 2: Figure 3 Reports the most common and frequent carcinogens in the workplace. Does the questionnaire include a question about the role of these subjects at their workplace?
Response 2: Thank you for your contribution. Yes, information about the function in the workplace and the activity carried out was used to report cases. This part has been added to the methods section (lines 321-324):
“The criterion used to notify cases were non-smoker, time of exposure to the carcinogen at work, latency time, and information about the function in the workplace and the activity carried out.”
Point 3: In order to assess the time of occupational exposure it is important to determine the types of job that occupation has. This simply means the people who actively work at the sites of productions or directly deal with the chemicals are likely to get high levels of exposure in comparison to the people who get passively exposed to the environmental toxicants generating through the chemicals which are being used in that occupation. Therefore, the intensity of exposure and its effects cannot be the same amongst the people in the same occupation. This information is very important to be considered to determine the time of exposure. Does the questionnaire include this? If not, it is important the add the question about the role of subjects/job profile in the questionnaire. And if questionnaire has the question, then this should be elaborate in the materials and methods parts of the article. If needed, the results section should also be modified after adding this information.
Response 3: We would like to thank for the comment and agreed that is important to measure the intensity of exposition. Howover, the main objective of this study was to create a simplified tool for collecting information on work environments and exposure to carcinogens, for routine use from a hospital with a large flow of patients, for epidemiological recording and possible notification of work-related cancer.
The focus was to identify potential occupational exposures that may have contributed to the incidence of cancer among patients, since the cancer outcome had already occurred. That is, to identify possible causal relationships between exposure to carcinogens at work and the cancer outcome.
The study identified 550 patients who had some type of exposure to carcinogens at work, but only 38 notifications were indicated with "possible" causal relationship to be investigated by the competent authorities. This suggests that the developed tool can be useful in identifying possible occupational exposures that may have contributed to the incidence of cancer among patients and can be used to generate new investigations.
Overall, the study highlights the importance of identifying and monitoring occupational exposures to carcinogens and developing effective tools to collect information about these exposures. These tools can help healthcare professionals identify possible occupational exposures that may have contributed to the incidence of cancer among patients, thus allowing for early detection and intervention.

Reviewer 2 Report
The authors are to congratulated for an online questionnaire to identify those cancer patients who have been occupationally exposed to carcinogens associated with their particular cancer.
The statement in the first sentence of the abstract that there has been little published on the methodology of questionnaire construction is not correct. There have been many such publications , which have generally involved the development and use of Job-Exposure Matrices, commonly referred to as JEMs.
The methodology for constructing the questionnaire is set out in five measures and consists of five measures, This methodology is sound, but there are some problems with each.
This It would be helpful if the authors were to state the criteria used to decide which cancers would be included in the list.
(i) For each type of cancer, list the carcinogens that could be present in the work place and afterwards. I presume this means to list all types of cancer that could be attributed to work-related exposure. The list does not include brain cancer, colorectal cancer or prostate cancer. There has been some published evidence linking each of these cancers with occupational exposures (although not necessarily conclusive evidence). There needs to be an explanation of the criteria used for what cancers are included and which are not.
(ii) For each type of cancer, list the carcinogens that could be present in the work place and afterwards. An explanation is needed as to how these carcinogens were assigned to particular cancers. In Figure 3 asbestos has been assigned to some cancers where no causal link has (to my knowledge) been identified. For example in Figure 3 asbestos is linked to nasal and paranasal cancer: asbestos has been linked to several cancers but not nasal and paranasal cancer. Similarly wood dust has been inexplicably assigned to several cancers (incorrectly) whereas it is missing from nasal and paranasal cancer where there is a known linkage. Overall the list of carcinogens is surprisingly small. Perhaps this a a reflection of a narrow industrial base of the Barretos district, but it seems that several important carcinogens are missing. I strongly recommend that the authors consult the literature on job-exposure matrices, such as those I have cited at the end of this review.
(iii) For each carcinogen, list the workplaces and / or professions in which they 130 could be present. It should be borne in mind that some occupational groups are at risk of excess cancers where no specific causal exposure has been identified, painting, firefighting.
(iv) Calculate the time of exposure. Presumably this means the duration of exposure.
(v) Calculate time of latency from exposure to onset of cancer. The questionnaire shown in Figure 2 assumes a latency period from the time exposure ceases until diagnosis. . Conventionally epidemiologists assume a latency period from the commencement of exposure until diagnosis. It is not possible to identify a time when a minimally-sufficient exposure has occurred to initiate the development of a cancer, but is certainly not appropriate to measure the latency period from the time exposure ceases.
In summary, this paper adequately describes the trial use of an online questionnaire, but the instrument itself requires improvement. The list of carcinogens and corresponding cancer sites appears to have been compiled without recourse to the large amount of research that has already taken place on the subject of job-exposure matrices and questionnaire design.
Suggested references
Kauppinen T, Uuksulainen S, Saalo A, Mäkinen I, Pukkala E. Use of the Finnish Information System on Occupational Exposure (FINJEM) in epidemiologic, surveillance, and other applications. Ann Occup Hyg. 2014 Apr;58(3):380-96. doi: 10.1093/annhyg/met074. Epub 2014 Jan 8. PMID: 24401793.
Lavoué J, Pintos J, Van Tongeren M, Kincl L, Richardson L, Kauppinen T, Cardis E, Siemiatycki J. Comparison of exposure estimates in the Finnish job-exposure matrix FINJEM with a JEM derived from expert assessments performed in Montreal. Occup Environ Med. 2012 Jul;69(7):465-71. doi: 10.1136/oemed-2011-100154. Epub 2012 Apr 1. PMID: 22467796.
Author Response
Response to Reviewer 2 Comments
Point 1: The statement in the first sentence of the abstract that there has been little published on the methodology of questionnaire construction is not correct. There have been many such publications , which have generally involved the development and use of Job-Exposure Matrices, commonly referred to as JEMs.
Response 1: Thank you for your contribution, the authors agree with you and the abstract changed to (lines 16-17):
“To obtain a history of occupational exposure in the workplace, the questionnaire is one of the main sources of information.”
Point 2: This It would be helpful if the authors were to state the criteria used to decide which cancers would be included in the list.
- For each type of cancer, list the carcinogens that could be present in the work place and afterwards. I presume this means to list all types of cancer that could be attributed to work-related exposure. The list does not include brain cancer, colorectal cancer or prostate cancer. There has been some published evidence linking each of these cancers with occupational exposures (although not necessarily conclusive evidence). There needs to be an explanation of the criteria used for what cancers are included and which are not.
Response 2 (i): The criteria for deciding which types of cancer to include in our questionnaire were based on the Work-Related Cancer Surveillance Guidelines[2] reported by the National Cancer Institute, where: "a literature review allowed us to identify the current state of the art for the characteristics of the main types of cancer, their epidemiological dimension and their risk factors. Briefly, at the end of each type of cancer, the main substances or physical and chemical agents (named in their set of agents), occupations and economic activities that pose a relevant risk for work-related cancer." “The text and tables use IARC monographs and an international review of the last 30 years carried out by Garbin (2010) based on works published in journals indexed in PubMed.”
Trerefore, as this questionnaire was based on the Work-related Cancer Surveillance Guidelines, the types of cancer included are those listed in:
"CHAPTER 3. TYPES OF CANCER AND RELATION TO OCCUPATIONAL EXPOSURE,
3.1 Skin
3.2 Lung
3.3 Mesothelioma
3.4 Bladder
3.5 Nasal, sinonasal, nasopharynx, oropharynx, larynx
3,6 Hematological (Leukemias, Multiple Myeloma, Non-Hodgkin's Lymphomas),
3.7 Stomach and esophagus
3.8 Liver
3.9 Pancreas,
3.10. Breast,
3.11. Brain and Central Nervous System".
These were the types of occupational cancers that we used in the questionnaire.
- For each type of cancer, list the carcinogens that could be present in the work place and afterwards. An explanation is needed as to how these carcinogens were assigned to particular cancers. In Figure 3 asbestos has been assigned to some cancers where no causal link has (to my knowledge) been identified. For example in Figure 3 asbestos is linked to nasal and paranasal cancer: asbestos has been linked to several cancers but not nasal and paranasal cancer. Similarly wood dust has been inexplicably assigned to several cancers (incorrectly) whereas it is missing from nasal and paranasal cancer where there is a known linkage. Overall the list of carcinogens is surprisingly small. Perhaps this a a reflection of a narrow industrial base of the Barretos district, but it seems that several important carcinogens are missing. I strongly recommend that the authors consult the literature on job-exposure matrices, such as those I have cited at the end of this review.
Response 2 (ii): All information for constructing the questionnaire was taken from the Work-Related Cancer Surveillance Guidelines
(https://www.inca.gov.br/sites/ufu.sti.inca.local/files//media/document//diretrizes-vigilancia-cancer-relacionado-2ed.compressed.pdf).
In the Work-Related Cancer Surveillance Guidelines, in chapter 3.5. Nasal, nasosinusal, nasopharynx, oropharynx, larynx, the following references and justifications for the inclusion of asbestos associated with nasal and paranasal cancer are listed:
"Exposure to formaldehyde increases the risk of developing adenocarcinomas and squamous cell carcinomas in men and women (Baan et al., 2009, Luce et al., 1992), but mainly of the nasopharynx (Laforest et al., 2000; Berrino et al., 1992). al., 2003). In addition, men with high levels of asbestos exposure have an increased risk of developing nasosinusal squamous cell carcinoma (Marsh et al., 2002; Luce et al., 2002)."
"Table 9. Risk factors for cancer of the nasal cavities and paranasal sinuses
AGENT: Chromium, nickel, cutting oil, wood dust, leather, cement, cereals and textiles, asbestos, formaldehyde, ionizing radiation, organochlorines, nickel and its compounds
OCCUPATION Carpenters and joiners, bakers (in general, in the chemical, coke and gas industries), miners, bricklayers, shoemakers, plumbers, car mechanics.
ECONOMIC ACTIVITY: Nickel foundry, industry: wood, production of isopropyl alcohol, leather and footwear, textiles, paper and oil, sawmill and carpentry, machine shop, foundry, agriculture.”
“Table 10. Risk factors for cancer of the oral cavity, pharynx and larynx
AGENT: Cutting oil, asbestos, dust from wood, leather, cement, cereals and textiles, asbestos, formaldehyde, silica, coal soot, organic solvents and pesticides, mist from strong acids
OCCUPATION: Hairdresser, carpenter, plumber, carpet installer, moulder, and glass shaper, potter, butcher and barber, miner and stonemason, painter, car mechanic
ECONOMIC ACTIVITY: Agriculture and animal husbandry; industries: textile, leather, metallurgy,
rubber, civil construction, machine shop, foundry, mining of coal.
Figure 3 represents our findings, in our patients, Barretos being a small city in the state of São Paulo, which is based on agriculture and livestock.
- For each carcinogen, list the workplaces and / or professions in which they 130 could be present. It should be borne in mind that some occupational groups are at risk of excess cancers where no specific causal exposure has been identified, painting, firefighting.
Response 2 (iii): We really didn't have cancer patients related to these professions. Most cancers were related to rural workers, which is a characteristic of this region of the country. However, these questions exist in the questionnaire and can be consulted. They will be inserted in the Supplementary Material. (SSQ_Attachment).
- Calculate the time of exposure. Presumably this means the duration of exposure.
Response 2 (iv): Yes, it calculates how long the patient has been in workplaces and/or professions where carcinogens may be present (Figure 2).
- Calculate time of latency from exposure to onset of cancer. The questionnaire shown in Figure 2 assumes a latency period from the time exposure ceases until diagnosis. Conventionally epidemiologists assume a latency period from the commencement of exposure until diagnosis. It is not possible to identify a time when a minimally-sufficient exposure has occurred to initiate the development of a cancer, but is certainly not appropriate to measure the latency period from the time exposure ceases.
Response 2 (v): Would like to thank the reviewer comment. This part has been added to the Materials and Methods, Figure 2 :
“The latency time was calculated using the exposure time of question 2, the time of question 3 and the date of the cancer diagnosis”
In summary, this paper adequately describes the trial use of an online questionnaire, but the instrument itself requires improvement. The list of carcinogens and corresponding cancer sites appears to have been compiled without recourse to the large amount of research that has already taken place on the subject of job-exposure matrices and questionnaire design.
I would like to thank the reviewer for the comment and add and justify that this study to construct the questionnarie was based on the already mentioned INCA Brazilian Guidelines with purpose to verify the feasibility of being used in the clinical routine to obtain information for purposes of epidemiological registration and possible compulsory notification of work-related cancer.

Reviewer 3 Report
This is a clear and (generally) well-written article, about an important subject, though of most interest for Brasilian readers. I have only a few minor requests for clarification.
1. line 225. What is "The notifiable term"?
2. Figure 2 (in fact really a table). Please indicate (e.g. in footnote) what is the base for the percentages? I.e. the denominators in the calculation
3. lines 273-274. The criterion used to notify cases ... (it should be criteria). Please be a bit more precise. What were the thresholds for time of exposure and latency time?
Author Response
Response to Reviewer 3 Comments
This is a clear and (generally) well-written article, about an important subject, though of most interest for Brasilian readers. I have only a few minor requests for clarification.
Point 1: Line 225. What is "The notifiable term"?
Response 1: It was inserted in the text, on page 6 (lines 236-238), highlighted in red:
“It refers to a term used in reporting occupational exposure that requires notification to the appropriate authorities. The “notifiable term” is used regardless of the duration of exposure or the time between exposure and the onset of symptoms.”
Point 2: Figure 2 (in fact really a Figure 3). Please indicate (e.g. in footnote) what is the base for the percentages? I.e. the denominators in the calculation
Response 2: It was inserted in the text, on page 10 (lines 314-315), highlighted in red:
“Figure 3: Distribution of the number and percentage of cancer patients who responded to SSQ and exposures to the most common and frequent carcinogens in the workplace.”
Point 3: lines 273-274. The criterion used to notify cases ... (it should be criteria). Please be a bit more precise. What were the thresholds for time of exposure and latency time?
Response 3: It was inserted in the text, on page 11 (lines 324-325), highlighted in red:
“The cases were analyzed individually and the minimum time of exposures and latency considered for reporting was 15 years.”

Round 2
Reviewer 1 Report
The suggestions have incorporated appropriately.
Author Response
Thank you for your considerations.
The English has been revised.
Reviewer 2 Report
The exposures, occupations and industries considered as potential causes of each cancer type are drawn from the Work-related Cancer Surveillance Guidelines published by the National Cancer Institute of Brazil.
Therefore the usefulness of the questionnaire will depend on the quality of Guidelines on which is based. It is therefore important to understand the criteria used by the National Cancer Institute when deciding which exposures are likely to cause a particular cancer. More specifically, what level of evidence is required to include a particular exposure (or occupation)? It has been difficult for this reviewer to answer this question as I do not speak Portuguese. From my reading of the Guidelines it appears that many of the exposures are included on the basis of very limited evidence.
It appears that the Guidelines are designed to ensure that no cases of occupational cancer are missed. Unfortunately, this means that many, if not most, of the cancers identified by the questionnaire will not be occupation-related. In other words, a high degree of sensitivity has been achieved at the cost of low specificity.
The most obvious example is the inclusion of agriculture and agricultural chemicals as a risk factor in many different cancer types. In their reply to me, the authors have pointed out that their study was undertaken in a rural area with a rural workforce, with the result that nearly one-half of the cancers are labelled as potentially occupation-related - mostly from agricultural employment and/or contact with pesticides - compared with the likely figure quoted by the authors of 4%.
Achieving a balance between specificity and sensitivity is difficult. My concern is not with the methodology of this study, but with the guidelines on which it is based.
The article will be of interest to people engaged in occupational cancer surveillance, but the tools used may not be very helpful in discriminating cancers that are occupationally caused from those that are not.
Author Response
Limitations of the study
Thank you for your considerations.
The English has been revised and the topic "Limitations of the study" has been included in the article (page 12, line 370-374) .
A limitation of the study is the high sensitivity and low specificity of the questionnaire. However, this questionnaire can signal occupational cancer surveillance about the main risk factors found in a population and steps can be taken to prevent, such as found in our population, farmers' exposure to agricultural chemicals.